

# EGFR mutation status yield from bronchoalveolar lavage in patients with primary pulmonary adenocarcinoma compared to a venous blood sample and tissue biopsy

Nikolay Yanev[1], Evgeni Mekov[1], Dinko Valev[2], Georgi Yankov[1], Vladimir Milanov[1], Stoyan Bichev[3], Natalia Gabrovska[1] and Dimitar Kostadinov[1]

[1] Department of Pulmonary Diseases, Medical University - Sofia, Sofia, Bulgaria
[2] University Hospital "St. Ioan Krustitel", Sofia, Bulgaria
[3] National Genetics Laboratory, Medical University - Sofia, Sofia, Bulgaria

Corresponding author
Evgeni Mekov, dr_mekov@abv.bg, evgeni.mekov@gmail.com

## ABSTRACT

**Background.** In recent years, there has been a revolution in the genomic profiling and molecular typing of lung cancer. A key oncogene is the epidermal growth factor receptor (EGFR). The gold standard for determining EGFR mutation status is tissue biopsy, where a histological specimen is taken by a bronchoscopic or surgical method (transbronchial biopsy, forceps biopsy, etc.). However, in clinical practice the tissue sample is often insufficient for morphological and molecular analysis. Bronchoalveolar lavage is a validated diagnostic method for pathogenic infections in the lower respiratory tract, yet its diagnostic value for oncogenic mutation testing in lung cancer has not been extensively investigated. This study aims to compare the prevalence of EGFR mutation status in bronchoalveolar lavage and peripheral blood referring to the gold standard - tissue biopsy in patients with primary lung adenocarcinoma.

**Methods.** Twenty-six patients with adenocarcinoma were examined for EGFR mutation from tissue biopsy, peripheral blood sample and bronchoalveolar lavage.

**Results.** Thirteen patients had wild type EGFR and the other 13 had EGFR mutation. EGFR mutation from a peripheral blood sample was identified in 38.5% (5/13) of patients, whereas EGFR mutation obtained from bronchoalveolar lavage (BAL) was identified in 92.3% (12/13). This study demonstrates that a liquid biopsy sample for EGFR status from BAL has a higher sensitivity compared to a venous blood sample.

## INTRODUCTION

In recent years, there has been a revolution in the genomic profiling and molecular typing of lung cancer. Full genomic profiling is recommended for the selection of the optimal personalized therapy.

The epidermal growth factor receptor (EGFR) is a key oncogene for targeted therapy along with anaplastic lymphoma kinase (ALK), v-Raf murine sarcoma viral oncogene homolog B (BRAF), MET, and ROS. Treatment with first and second-generation EGFR tyrosine kinase inhibitors such as erlotinib, gefitinib, and afatinib is the standard of care in patients with sensitive mutations (Exon 19 Deletions, L858R) in the kinase domain of the EGFR gene (*Novello et al., 2016*).

The gold standard for determining the EGFR mutation status is tissue biopsy, where a histological specimen is taken by a bronchoscopic or surgical method (transbronchial biopsy, forceps biopsy, etc.). However, in clinical practice the tissue sample is often insufficient for morphological and molecular analysis (*Coghlin et al., 2010*; *Rivera, Mehta & Wahidi, 2013*; *Weber et al., 2014*).

In approximately 80% of non-small cell lung cancer patients, the only sample available is either cytological material or a limited amount of tissue (*Bubendorf et al., 2017*). What is more, a biopsy is an invasive procedure associated with a higher risk for complications (*Lorenz & Blum, 2006*) and the results of genetic biopsy examinations of a biopsy sample from a single tumor site is not a reflection of tumor heterogeneity. Hence, the clinician is unable to observe the molecular resistance in the treatment process (*Gerlinger et al., 2012*). On the other hand, the isolation of free circulating peripheral blood tumor DNA (ctDNA) known as 'liquid biopsy' is a non-invasive, alternative method to tissue biopsy for performing molecular screening in patients suitable for targeted therapy. The sensitivity and specificity of the method vary depending on the research platform and the quality of the test material (*Sacher, Komatsubara & Oxnard, 2017*).

This study aims to compare the prevalence of EGFR mutation status in bronchoalveolar lavage samples with the prevalence in peripheral blood, referring to the gold standard—tissue biopsy, in patients with primary lung adenocarcinoma.

## MATERIAL AND METHODS

We assessed all patients scheduled for bronchoscopy in our hospital between October 2018 and August 2019. The patients were further evaluated according to the protocol if lung adenocarcinoma was confirmed. The exclusion criteria were in concordance with the standard contraindications for a bronchoscopy. The research protocol was approved by the Ethical Committee of the Medical University Sofia (approval letter 1601/15.04.2019). Written informed consent was obtained from all patients included in this study before they underwent bronchoscopy.

Flexible bronchoscopy was performed in all patients under local anesthesia with lidocaine (10% lidocaine spray) in nasal passages, followed by nasal instillation of a 2% solution during the procedure through the working channel, achieving a total dose of up to 8 mg/kg. The bronchoscopies were performed with Olympus Exera II CV-180, Pentax EPK-1000 and Fujinon EB-530T optical and video devices. The biopsy techniques were forceps biopsy, transbronchial biopsy with or without fluorographic control, based on preliminary CT data, and bronchoalveolar lavage (BAL).

BAL was performed under local anesthesia after an examination of the tracheobronchial tree. The 0.9% saline, which was used for the procedure, was preheated to body temperature.

The fluid was instilled with syringes through the biopsy canal by repeatedly injecting 20–60 ml (usually four or five times) to a total volume of 100–240 ml. After each administration the fluid was aspirated back and the aspirated volume amounted to 40–70% of the initially administered one.

Obtained biopsy materials were labeled and transported immediately to the Department of Clinical Pathology, with histological materials fixed in formalin solution, whereas the cytological materials were applied to a glass slide—a total of four pieces. BAL and plasma samples were frozen immediately at −10 °C. No BAL sample underwent centrifugation.

The histological materials were evaluated by a clinical pathologist through via light microscopy and stained with hematoxylin-eosin. Additional immunohistochemistry and genetic tests were performed if necessary.

In cases where adenocarcinoma was confirmed from a tissue sample, the frozen plasma and BAL samples were sent for DNA analysis in the National Genetic Laboratory. There, DNA extraction was performed with QiaAmp Circulating Nucleic Acid kit for plasma and BAL samples. For Formalin-Fixed Paraffin-Embedded Tissue (FFPET) samples, we used the QIAamp DNA FFPE Tissue Kit. EGFR genotyping was performed via PCR with therascreen EGFR Plasma or tissue RGQ PCR Kits provided by QIAGEN®. The platform used was the Rotor-Gene Q PCR. Prior to the beginning of the project, a validation of the assay for lavage samples was conducted. DNA was extracted from six BAL samples of patients with an already determined EGFR status. DNA extraction was performed with the Qiavac24 system. Four of the patients had wtEGFR and from the remaining two, one was positive for an Exon 19 deletion and one for L858R point mutation. The obtained DNA concentration was in the range of 7.5–25 ng/µl. PCR analysis of the samples was performed according to the IFU of the therascreen® EGFR Plasma RGQ PCR Kit Handbook (January 2019). All samples were amplified in three receptions. EGFR mutations were confirmed in both samples and in all replicates. Detected Ct and delta Ct values were in the range described in the manual.

## RESULTS

Between October 2018 and August 2019, 140 bronchoscopies were performed. In 112 patients (80%) the histopathological analysis of the acquired tissue samples confirmed the presence of a lung malignancy (Table 1). Due to the limited size of the patient sample, the prevalence of adenocarcinoma and EGFR mutations deviates to a certain extent from established data. Adenocarcinoma was present in 23.2% (26/112) of the patients with lung cancer which was verified through immunohistochemistry on tissue samples. Thirteen patients were female and 13 were male with a mean age of 63.3 years. The localization of the tumor was evenly spread with 13 left-sided, 12 right-sided, and one bilateral (without distinctive primary lesion) cases. The average maximum diameter of the tumors was 5.53 cm (min 2.3 cm, max 12.4 cm). Interestingly, a visible, intralumenal tumor was present in 23/26 patients (88.5%).

The groups did not differ with respect to the age and tumor maximum diameter (Table 2). However, there were significantly more women with EGFR mutation (77%, 10/13) as opposed to men (23%, 3/13, $p = 0.017$).

**Table 1  Patient demographics.**

| Characteristic | Value |
|---|---|
| Age, years, mean (±SD) | 64.4 (±8.4) |
| Male gender, *n* (%) | 100 (71.4) |
| **Histology** | ***n*** (%) |
| – Adenocarcinoma | 26 (18.6) |
| – Squamous cell carcinoma | 36 (25.7) |
| – Non-squamous cell carcinoma | 23 (16.4) |
| – Small cell carcinoma | 27 (19.3) |
| – Chronic nonspecific inflammation | 18 (12.9) |
| – Unidentified lung disease | 10 (7.1) |

**Table 2  Patient demographics and tumor characteristics.**

| Characteristic | EGFR mutation group (*n* = 13) | Wild type EGFR group (*n* = 13) |
|---|---|---|
| Mean age, years | 62 | 64.5 |
| Male gender, n | 3 | 10 |
| Tumor localization | | |
| – Left (n) | 7 | 6 |
| – Right (n) | 5 | 7 |
| – Both sides (n) | 1 | 0 |
| Tumor diameter, mean (cm) | 4.9 | 5.9 |
| Visible tumor (n) | 10 | 13 |

Thirteen patients had wild type EGFR and the other 13 had EGFR mutation. EGFR mutation from a peripheral blood sample was identified in 38.5% (5/13) patients, whereas EGFR mutation obtained from BAL was identified in 92.3% (12/13) (Table 3). In one case in both liquid biopsy samples (BAL and plasma) an additional T790M mutation was detected, along with a deletion in exon 19, which was not established in the corresponding FFPET sample (patient 8). Neither BAL, not blood sample confirmed EGFR mutation in 1 patient which is positive only from a tissue sample (Table 4).

## DISCUSSION

The present study compares the detectability of activating EGFR mutations in BAL and blood plasma samples, compared to the result from the tissue biopsy. In this study, BAL was found to be more sensitive with regards to detecting an EGFR mutation compared to liquid biopsy –92.3% vs. 38.5% in the same patients. This result is consistent with the published data in other studies (*Park et al., 2017*). Moreover, in two cases, BAL provides additional data about the mutational status, compared to histology specimens (Table 4, case numbers 8, and 12), a fact most likely attributable to tumor heterogeneity and tissue sampling process. Thus, more information and more treatment choices may be available for the patients.

**Table 3  Confirmation of EGFR mutation in BAL and blood samples.**

|  | Confirmed EGFR mutation |
| --- | --- |
| BAL sample | 12 (92.3%) |
| Blood plasma | 5 (38.5%) |
| Total patients | 13 (100%) |

**Table 4  EGFR mutation status in the individuals.**

| N | FFPET | BAL | Blood plasma |
| --- | --- | --- | --- |
| 1 | Exon 19 deletion | Exon 19 deletion | Wild type |
| 2 | Exon 21 L858R | Exon 21 L858R | Wild type |
| 3 | Exon 21 L858R | Wild type | Wild type |
| 4 | Exon 19 deletion | Exon 19 deletion | Wild type |
| 5 | Exon 21 L858R | Exon 21 L858R | Wild type |
| 6 | Exon18 G719 | Exon 18 G719 | Exon 18 G719 |
| 7 | Exon 19 deletion | Exon 19 deletion | Exon 19 deletion |
| 8 | Exon 19 deletion | Exon 19 deletion/T790M[*] | Exon 19 deletion |
| 9 | Exon 19 deletion | Exon 19 deletion | Exon 19 deletion |
| 10 | Exon 19 deletion | Exon 19 deletion | Wild type |
| 11 | Exon 19 deletion | Exon 19 deletion | Exon 19 deletion |
| 12 | Exon 21 L858R | Exon 21 L858R/Exon 19 deletion[*] | Wild type |
| 13 | Exon 19 deletion | Exon 19 deletion | Wild type |

**Notes.**
[*]Additional mutation.

The rapid growth of tumor cells leads to apoptosis and necrosis. In some cases, the amount of post-apoptotic residues and fragments exceeds phagocytic clearance. This leads to the presence of a certain concentration of tumor cell debris, including ctDNA, into the systemic circulation, where they are isolated from *Diehl et al. (2008)*. The use of blood plasma for the detection of ctDNA is a non-invasive method, which represents a feasible option in cases where tissue biopsy is not possible. However, it requires the use of more sensitive diagnostic methods due to the very small amount of ctDNA in plasma (*Zachariah et al., 2008*; *Zanetti-Dallenbach et al., 2007*). The results from a meta-analysis of 3110 patients showed that the blood plasma liquid biopsy has 62% sensitivity and specificity of 95%. Although the sensitivity is lower in comparison with tissue biopsy, the authors state that ctDNA testing is an effective diagnostic method for EGFR mutation detection with a very high specificity (*Qiu et al., 2014*).

A limited number of studies have been directed to the study of EGFR mutational status from bronchoalveolar lavage. Bronchoalveolar lavage has some advantages - a semi-invasive method associated with a lower risk complications due to the procedure, compared to tissue biopsy (*Herth, Shah & Gompelmann, 2017*), even at more advanced stages of the disease. Furthermore, since more cellular debris, resulting from tumor apoptosis and necrosis, accumulate in a localized area in the lungs before entering the vascular system, the ctDNA concentration would be higher in bronchoalveolar lavage (BAL) than in plasma (*Jahr et*

*al., 2001*; *Stroun et al, 2001*). Hence, one could make an assumption that the probability to identify the EGFR mutation status would be higher.

Also, in bronchoalveolar lavage, it is possible to have ctDNA fragments from different regions of the tumor surface (tumor heterogeneity), which increases the chance of identifying mutations from multiple tumor locations.

*Schmidt et al. (2004)* isolated free tumor lavage DNA in 46.7% (14/30) of patients. In *Park et al. (2017)*, 91.7% of BAL established EGFR status compared to tumor biopsy using a high-sensitivity method.

Currently, re-biopsy is the gold standard for the detection of EGFR mutations in patients with unknown status after negative tissue sampling. Unfortunately, re-biopsy is not always feasible due to many reasons (comorbidity and condition of the patient, difficulties in obtaining biopsy specimens). In this context, liquid biopsy could be an effective solution. Results from individual studies are variable, and even though its sensitivity is approximately 62% (*Qiu et al., 2014*), ctDNA is a very specific and effective biomarker. Furthermore, establishing EGFR status from BAL is a viable and safe procedure as well, especially in impaired patients with promising results. Since the BAL samples didn't undergo centrifugation, one could not say for sure if the positive result derives from ctDNA or from tumor cells, present in the sample. Lastly, one should not underestimate the fact that 88.5% of the patients with adenocarcinoma (23/26) had visible intralumenal tumors. The correlation of the BAL sensitivity with the presence of intralumenal lesions needs further investigating, as BAL could turn out to be a workable diagnostic tool in such clinical cases.

## CONCLUSION

We managed to demonstrate a higher sensitivity of BAL to venous blood liquid biopsy in establishing an EGFR mutation in the patient population, which participated in this study. This could facilitate a diagnosis with a minimally invasive method and lower risk for the patient with comparable results to FFPET in this limited group of patients.

### Funding

This work was supported by the Medical University—Sofia Grant D-72/23.04.2019. The funders had no role in study design, data collection and analysis, decision to publish, or preparation of the manuscript.

### Grant Disclosures

The following grant information was disclosed by the authors:
Medical University—Sofia: D-72/23.04.2019.

### Competing Interests

The authors declare there are no competing interests.

## Author Contributions

- Nikolay Yanev and Stoyan Bichev conceived and designed the experiments, performed the experiments, analyzed the data, prepared figures and/or tables, authored or reviewed drafts of the paper, and approved the final draft.
- Evgeni Mekov conceived and designed the experiments, analyzed the data, prepared figures and/or tables, authored or reviewed drafts of the paper, and approved the final draft.
- Dinko Valev and Dimitar Kostadinov conceived and designed the experiments, performed the experiments, authored or reviewed drafts of the paper, and approved the final draft.
- Georgi Yankov, Vladimir Milanov and Natalia Gabrovska conceived and designed the experiments, authored or reviewed drafts of the paper, and approved the final draft.

## Human Ethics

The following information was supplied relating to ethical approvals (i.e., approving body and any reference numbers):

The research protocol was approved by the Ethical Committee of the Medical University Sofia. (approval letter 1601/15.04.2019).

## Data Availability

The raw data are available in the Supplemental File.

## Supplemental Information

Supplemental information for this article can be found online at http://dx.doi.org/10.7717/peerj.11448#supplemental-information.

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
