# Peer review of "EGFR mutation status yield from bronchoalveolar lavage in patients with primary pulmonary adenocarcinoma compared to a venous blood sample and tissue biopsy"

_PeerJ, doi:10.7717/peerj.11448_

## Round 0.1 · original submission · Major Revisions

Dear authors,

Your manuscript has now been reviewed by three reviewers.
As you will see, although they find your manuscript interesting, they have raised a large number of concerns, implying a thorough revision and reformulation of the work.

Please address ALL the concerns raised by the Reviewers, in a point-by-point rebuttal letter, in order to be possible to consider your work further.

My best regards

Reviewer 1 ·

Basic reporting

Nikolay et al. reported the testing results from bronchoalveolar lavage potentially be used for EGFR mutation status testing. The conclusion of this manuscript is liquid biopsy sample for EGFR status from BAL is superior to a venous blood sample. For this work, I mainly have some questions and comments as follows.

Experimental design

1. In this study, why did the authors choose serum samples instead of plasma samples to analyze the EGFR mutation status? The serum sample contains a good deal of cfDNA, it affect the testing rate.
2. Which platform their used to test the EGFR status for plasma and BAL samples? They should describe it in the method section.

Validity of the findings

No comments.

Additional comments

The authors should reorganize the study design and data, and discuss the advantages and disadvantages of the three approaches for EGFR status testing, rather than simply draw a conclusion that the BAL testing is superior to the blood testing.

Reviewer 2 ·

Basic reporting

The article is written in English language that is unambiguous and rather clear. Please consider a minor revision.
Regarding, the background described by authors introduces well into the context of the study. The urge of using liquid biopsies instead or as a surrogate of tissue biopsies is well justified. I am however missing a one-two sentence link to the use of BAL specimen. In addition, I am missing a mention of reports on the current knowledge. Regarding EGFR detection in blood and BAL – especially that this study is not novel in that sense. More emphasis should be put on the fact that this is PCR not NGS method.
Most of the references cited for this study is relevant. Reference [7] cited is a little obsolete: the methods of EGFR detection cannot be compared directly, moreover there is also no clarity regarding the material used (ctDNa from BAL supernatants or whole BAL sample). Once clarified the issue from pt. 1, please go back to the references and discuss accordingly.
Please revise Reference [14] for its details that appear in PubMed. Found only this: Carstensen T, Schmidt B, Engel E, Jandrig B, Witt C, Fleischhacker M. Detection of cell-free DNA in bronchial lavage fluid supernatants of patients with lung cancer. Ann N Y Acad Sci. 2004 Jun;1022:202-10. doi: 10.1196/annals.1318.031. PMID: 15251961.

Experimental design

The most important issue to comment is that it is not entirely clear if the authors worked with ctDNA from BAL (supernatant) or DNA isolated from the cellular debris. Based on the kit used (Circula, I assume the first. If the BAL was not processed via centrifugation (if was – please provide info about the speed, time etc) – we can hardly agree on the fact that it was ctDNA – it could derive from the cells that are present in the BAL. Although this detail does not impair the utility of BAL as a diagnostic tool, this point merits certain clarification.
In Materials and Methods please add the provider of all the kits. In addition, for more clarity I would suggest to move the paragraph regarding Ethical Committee where patient recruitment. I suggest dividing this part into 1) Patients 2) Samples collection and processing 3) DNA extraction and genotyping.

The study is missing table with patient characteristics.
Reg. Table 2 I suggest changing the Legend. Either mark with “*” only the novel confirmed mutations (pts 8 & 12) or add another mark for these additional mutations detected in blood/BAL. This is an important information and gets lost in the Table.

Validity of the findings

This study is not novel in the field. It does provide, however, meaningful confirmation of the utility of BAL for EGFR detection.

Would be very interesting to see the possibility of detection of a panel of mutations in these samples, however it is understandable that the clinical utility of such study must come in line. Namely, if there is no centrally refunded therapies targeted in other drivers that EGFR in Bulgaria, probably has not much sense to be performed today, but could serve as a good premise to include other targeted therapies into clinical practice. Maybe you could speculate on that in a one-sentence more of the discussion.

Reviewer 3 ·

Basic reporting

Basic Reporting:
In general, the manuscript is well written considering the authors are from Bulgaria. Context is easily understandable, but would benefit from a professional medical writer fluent in English to clean up the paper. The text is technically correct.
The Introduction and background are sufficient for the content of the paper. The comment on lines 50-51 regarding the difficulty of obtaining sufficient tumor tissue needs to be referenced such as in the paper by Wu et al (Lung Cancer, 2018, Vol 126, pp 1-8). On line 54, it is unclear how reference 3 relates to complications of a tumor biopsy, could a better reference be found?
The structure of the article is consistent with accepted norms. There are only two tables which are relevant to the content of the article. Raw data in the form of an excel file of EGFR mutations which is germane to the content of the article. There should be a demographic table on the patients in the study, including age, sex, performance status if available, smoking status and stage.
The article appears to be self-contained and a complete body of work and not subdivided from a larger body of work.

Experimental design

Experimental design
The article is within the scope of the journal related to the medical sciences. The research question is adequately defined and meaningful for patients with lung cancer. The authors define a medical need and the study contributes to bridging the medical need. The investigation appeared to be conducted in accordance with accepted norms. On lines 94-95 the authors state that the protocol was approved by an ethics committee and informed consent was obtained. However, it is not clear when informed consent was obtained. Was it obtained prior to the initial bronchoscopy or after lung adenocarcinoma was confirmed (line 71)? Additionally, how was the lung cancer confirmed? Was it by a tissue biopsy, cytology or something else? What was the gold standard used in this study?
The flexible bronchoscopy is described in reasonable detail (lines 73-79). However, the broncho-alveolar lavage (BAL) (line 79) is not well-described and should be expanded upon. What type of fluid was used and how much? Was there a standard recovery? If the technique used is detailed elsewhere, it should be referenced. However, information on the amount of lavage fluid recovered would be interesting to include in the article.
The authors used the therascreen® EGFR Plasma RGQ PCR Kit for testing BAL specimens. This polymerase chain reaction (PCR) assay has been validated in plasma (therascreen® EGFR Plasma RGQ PCR Kit Handbook, January 2019) and not in lavage samples. Did the authors conduct their own laboratory validation of the assay, as Ct cutoffs for PCR assays usually vary be sample type. Please describe any validation performed.

The authors list 112 patients with a “tumor” (line 99) but do not specify what this means. Were they malignant? Were they adenocarcinomas? They should expand here. A table with the full 140 bronchoscopies listing sex, age, maybe smoking status, tumor type (adenocarcinoma, squamous cell, etc.) It is also stated that only 23.2% (26) of the 112 tumors were adenocarcinomas which seems unusual without knowing what the others were. Among 112 primary lung malignancies, one would expect at least 60% to be adenocarcinomas or non-squamous cancers. Please explain this.

The authors state that among the 26 adenocarcinomas, there were 13 that were EGFR positive and 13 that were EGFR negative. This proportion (50%) is too high for a European population where one would expect an EGFR positivity rate of 10-15% (Lancet Oncology, 2012, Vol 13, pp239-46) as was seen in the EURTAC trial (14%). Please reconcile this. Were there actually more than 26 adenocarcinomas and 13 were EGFR positive and the 13 EGFR negative patients were chosen from the remaining adenocarcinomas? If so, how were these 13 chosen?

Validity of the findings

Validity of findings.

The authors present their data in a straightforward manner using the tissue biopsy EGFR results as the truth set which seems appropriate. Underlying data was provided using a commercially available EGFR assay.

The conclusions need some editing. The comment (line124) needs context. EGFR testing by BAL may be better (not superior) to plasma EGFR testing in certain situations. The authors failed to emphasize the most interesting finding in that 23/26 of the patients had visible intra-lumenal tumor. They should include how many of the EGFR positive samples had visible. In this context, EGFR testing may be better than plasma EGFR testing in patients with visible intralumenal tumors.

In the same first paragraph of the discussion, the authors state that the BAL results provided additional mutation data in cases 6,8 and 12. Was there any consideration to do orthogonal testing with NGS to confirm discordant results?

In the second paragraph of the discussion (lines 131-139), the authors discuss the low sensitivity of plasma EGFR testing compared to histological samples and do not recommend plasma testing. Commercially available PCR based, plasma EGFR assays (cobas EGFR Mutation Test and the therascreen EGFR assay) clearly state that plasma testing is best suited for those patients in whom a tissue sample is not available. Furthermore, the sensitivity and specificity of a plasma test is dependent on which EGFR mutation is tested and also the patient’s tumor burden. Thress et al (Lung Cancer, 2015, vol 90, pp 509-15) show much better sensitivity and specificity for exon 19 deletions and L858R mutations for both the cobas and therascreen assays than is stated in the paper (line 137). Wu et al (Lung Cancer, 2018, Vol 126, pp 1-8), showed that the rate of plasma EGFR positivity was correlated with tumor burden. The authors should expand on this paragraph for better context. As they have the tumor diameters for 17 of the 26 samples, did they look at those patients who were plasma positive based on the size of the tumor? It is fair to say that plasma EGFR testing is less sensitive, but as the thrust of this paper is about EGFR positivity in BAL, the statement regarding not recommending plasma EGFR testing (line 137-138) should be deleted. Just stating that plasma testing is less sensitive than tissue testing should be sufficient here.

The authors state that BAL is a semi-invasive method with a lower risk for the patient (line 142). Lower risk than what? Tissue sampling? If so, please provide references that support this contention.

In lines 163-164, the authors should state that BAL can be an alternative method to establish EGFR mutational status, especially in patients with visible intralumenal tumor. Again, the most interesting finding was that 90% of patients had visible tumor and I would emphasize that consideration of BAL for EGFR testing should be considered in this situation.

The conclusion is over-stated. BAL may be better than plasma EGFR testing in certain situations and provide valuable information for the treating physician.

Additional comments

General comments:

The information presented in this paper is interesting and adds to a relatively small body of work on using BAL to perform EGFR testing. Most of the comments should be easy to adjudicate. I appreciate the opportunity to evaluate the manuscript.

---

## Round 0.2 · Minor Revisions

Thank you for your modifications to the Ms. Although the reviewers found the manuscript is improved, there remain some points that need your attention.

Please address all the Reviewers' suggestions. Note that you have an annotated PDF with suggestions from Reviewer 3.

Reviewer 2 ·

Basic reporting

Minor revision of English has been performed. Authors have referred to comments regarding the references accordingly.
I have no new comments to make on that section.

Experimental design

I am mostly satisfied with Authors' comments and changes.

Please only make sure not to use the therms "liquid biopsy" and "ctDNA" in an abiguous manner. See lines 158, 172/173 of the tracked changes document, where "liquid biopsy" is once used as referred to both plasma and BAL (which I find correct), but later it refers to plasma only (172/173).

Please do not use the therm "ctDNA" when referring to tissue biopsy (line 221 of the tracked changes document), as well as whenever discussed regarding BAL samples analyzed in this study.

Validity of the findings

I am mostly satisfied with Authors' comments and changes.

Additional comments

Thank you for your revision. I find this study almost ready to be published.

Reviewer 3 ·

Basic reporting

The authors have addressed most of the comments and the language has been updated. I have made some minor edits to improve readability and will include the word document with my submission.
The addition of the demographic tables is welcome, but one glaring omission is the stage of the patients tested. This information may help explain why the plasma sensitivity was so low. Additionally, it is currently recommended that patients with non-squamous NSCLC should undergo molecular testing, so it is unclear why the 23 patients with non-squamous histology were not included.

Experimental design

The authors have done a good job of answering my comments.
After seeing the Table 1, the proportion of patients with squamous, non-squamous and small cell seem more in keeping with what would be expected. Also, if you include the non-squamous patients, then perhaps the proportion of EGFR positive patients would be more in line with what would be expected. However, it is not clear why the authors did not include all patients with non-squamous NSCLC in their analysis and limited it to those with a diagnosis of adenocarcinoma as current recommendations are that all patients with non-squamous NSCLC should undergo molecular testing for EGFR.

Validity of the findings

Overall, the authors have again done a good job in answering my concerns. However, one glaring omission remains. The authors still insist on using the reference by Li concerning a meta-analysis of published literature. The paper is quite old (2014) and reports on literature published between 2005 and 2013, well before the current commercial validated approved assays were available. They should find an updated reference. Their comment on lines 190-193 that use of plasma is not a feasible alternative to tissue needs to be put in context or modified. In advanced NSCLC, the sensitivity of plasma testing for EGFR mutations can range from 60-70% for T790M up to 85-90% for exon 19 deletions again referencing the Thress and Wu papers in the previous review. Also, with today’s modern assays, specificity in plasma for EGFR is in the 95-99% range. There may be good reasons for the low sensitivity in this study such as Stage of disease. Earlier stage disease is less likely to be ctDNA positive in plasma than advanced metastatic disease where most of the available data comes from. Stage is correlated with tumor burden to some degree, but as the authors have demographic data, Stage should be included. It would be more reasonable to say that testing for EGFR mutations in BAL may be better than plasma in certain situations, like in patients with visible intra-luminal disease. However, in advanced non-squamous NSCLC it is certainly feasible to detect EGFR mutations with acceptable sensitivity in patients who have insufficient tissue for molecular testing or cannot undergo an invasive biopsy. The authors may want to put a line or two in speculating why the plasma positive rate was low, including the fact that this was a very limited group of patients.

Additional comments

The authors have done a good job of answering the comments and outside of the comments above feel it is acceptable for publication.

Annotated reviews are not available for download in order to protect the identity of reviewers who chose to remain anonymous.

---

## Round 0.3 · accepted · Accept

Dear Authors,

Thank you for the modifications done in the manuscript, that have addressed the Reviewers concerns.

Your manuscript is now acceptable for publication.

As a minor correction please delete the word AIMS as a heading in line 71.

Best regards